# Being Stung Once or Twice by Bees (*Apis mellifera* L.) Slightly Disturbed the Serum Metabolome of SD Rats to a Similar Extent

**DOI:** 10.3390/ijms25126365

**Published:** 2024-06-08

**Authors:** Xinyu Wang, Xing Zheng, Xue Wang, Quanzhi Ji, Wenjun Peng, Zhenxing Liu, Yazhou Zhao

**Affiliations:** State Key Laboratory of Resource Insects, Institute of Apicultural Research, Chinese Academy of Agricultural Sciences, Beijing 100093, China; 82101235491@caas.cn (X.W.); zhengxing@caas.cn (X.Z.); wangxue41996@163.com (X.W.); 18336369851@163.com (Q.J.); pengwenjun@vip.sina.com (W.P.)

**Keywords:** honeybee sting, SD rats, ^1^H NMR, serum metabolome

## Abstract

In most cases, the number of honeybee stings received by the body is generally small, but honeybee stings can still cause serious allergic reactions. This study fully simulated bee stings under natural conditions and used ^1^H Nuclear Magnetic Resonance (^1^H NMR) to analyze the changes in the serum metabolome of Sprague–Dawley (SD) rats stung once or twice by honeybees to verify the impact of this mild sting on the body and its underlying mechanism. The differentially abundant metabolites between the blank control rats and the rats stung by honeybees included four amino acids (aspartate, glutamate, glutamine, and valine) and four organic acids (ascorbic acid, lactate, malate, and pyruvate). There was no separation between the sting groups, indicating that the impact of stinging once or twice on the serum metabolome was similar. Using the Principal Component Discriminant Analysis ( PCA-DA) and Variable Importance in Projection (VIP) methods, glucose, lactate, and pyruvate were identified to help distinguish between sting groups and non-sting groups. Metabolic pathway analysis revealed that four metabolic pathways, namely, the tricarboxylic acid cycle, pyruvate metabolism, glutamate metabolism, and alanine, aspartate, and glutamate metabolism, were significantly affected by bee stings. The above results can provide a theoretical basis for future epidemiological studies of bee stings and medical treatment of patients stung by honeybees.

## 1. Introduction

In many parts of the world, poisonous animals are a serious threat to human life and health, especially in Africa, Southeast Asia, and Central America. Epidemiological research on poisonous animals has mostly focused on snake venom or scorpion venom [1,2,3], and there is less epidemiological information on cases of bee stings. Limited recent research has shown that the frequency of bee stings is influenced by various factors, including the type of bee species, climate, environment, biodiversity level, population density, economic activities, and type of residence in different countries or regions [4]. Bee stings may cause two distinct clinical manifestations that primarily depend on the patient’s sensitivity to bee venom and the number of stings. These toxic reactions are attributed to the pharmacological activity of bee venom, and allergic reactions are due to the sensitization mechanism [5]. Most people experience local pain, edema, and itching after being stung by a small number of bees, and systemic reactions are relatively rare. This indicates that bee stings mainly cause harm to humans through the action of toxic substances in bee venom on blood vessels rather than through allergic mechanisms. Bee stings that lead to rare deaths are mostly caused by many bee stings or the patient’s sensitivity to bee stings [6]. However, the time from bee stings to death is more than 3 h in all cases, indicating that timely medical treatment is an important means to avoid tragedies. Therefore, epidemiological investigations of bee sting events have focused on individual testing for sensitization to bee stings, which can help to make more reasonable evaluations and provide information for future patient diagnosis and treatment.

Bee venom contains allergens and toxins that can cause allergic or toxic reactions in the body [7,8]. Allergic reactions are relatively easy to treat, but toxicity can lead to severe acute renal failure, liver failure, multiple organ failure, and even death [9,10]. The analysis of serum metabolomics in patients stung by bees may provide new insights into the pathogenic mechanism of bee stings and provide a theoretical basis for reducing multiple organ failure and death in clinical practice. Bee stings have a significant impact on the life systems of the body, leading to substantial changes in the metabolism of patients stung by bees. In principle, these changes can be revealed by metabolomics. The purpose of metabolomics is to qualitatively and quantitatively analyze all metabolites in biological systems (cells, tissues, organs, and organisms) under defined conditions, including the dynamic metabolic response of the life system to certain biological stimuli [11]. Matysiak et al. evaluated the response of the body to bee stings by analyzing the serum levels of 34 free amino acids [12]. Serum samples were collected from beekeepers 3 h and 6 weeks after bee stings, and significant differences in the serum concentrations of L-glutamine, L-glutamate, L-methionine, and 3-methyl-L-histidine were detected. In addition, L-glutamine and L-glutamate were found to be potential markers for bee stings and can be used to assess the recovery level of the body after bee stings. According to medical research reports, patients with severe allergic symptoms after being stung are generally treated with blood dialysis and can recover within 72 h. The metabolites in patient serum are correlated with disease symptoms and can be used to assess later recovery or treatment effects. Alonezi et al. studied the effect of melittin on the metabolism of ovarian cancer cells and demonstrated that melittin affects multiple metabolic pathways, such as the tricarboxylic acid cycle and oxidative phosphorylation [13]. To date, the changes in serum metabolites after bee stings in humans have not been fully elucidated. Arjmand et al. used proton nuclear magnetic resonance technology to study the mechanism underlying the toxicity of scorpion (*Hemiscorpius lepturus*) venom [14]. By analyzing the effects of venom on biochemical pathways and physiological responses in the body, they found that the most affected metabolic pathways were pyrimidine, histidine, and tyrosine metabolism and steroid hormone biosynthesis. Additionally, crude scorpion venom affects the pancreas and spleen of the body, and neural cells are attacked. The external manifestations are similar to the early symptoms of myocardial injury and epilepsy.

In most cases, the number of stings received by the body is generally small, that is, stung once or twice, and the body is very sensitive to such a small number of stings. The symptoms range from mild local inflammatory reactions to severe allergic reactions, even anaphylactic shock [15,16,17]. In addition, animals have the innate characteristics of seeking benefits and avoiding harm and rarely suffer from many stings. Therefore, we fully simulated the natural conditions of bee stings and analyzed the changes in the serum metabolome of SD rats stung once or twice by bees to verify the impact of such mild stings on the body and reveal the underlying mechanisms, which have practical significance.

## 2. Results

### 2.1. Analysis of Serum Metabolites in SD Rats Stung by Bees

Figure 1 shows the 600 MHz ^1^H NMR spectra of the serum metabolome of SD rats stung one time by bees. The spectra were typical and represented the ^1^H NMR spectra of all treatment groups and control groups. There were obvious differences in the contours between them. The resonance spectra of the metabolites were matched with the Chenomx database, and the given concentration of the internal standard DSS was used as the benchmark. By using Chenomx software (version 8.0, Chenomx, Edmonton, AB, Canada), 22 amino acids and their derivatives and 20 organic acids were identified qualitatively and quantitatively. The specific results are shown in Table 1 and Table 2. The levels of the amino acids aspartate, glutamate, glutamine, and valine and the organic acids ascorbate, lactate, malonate, and pyruvate significantly differed (*p* < 0.05) between the serum of the SD rats in the NC group and the treated groups (SO and ST), but no compounds differed between the treated groups (SO and ST). These findings indicate that the serum metabolites of SD rats stung by bees were significantly affected, but there was no significant difference in the serum metabolites between the SO group and the ST group.

### 2.2. Effect of Bee Sting Frequency on Serum Metabolites in SD Rats

Principal component discrimination analysis (PCA-DA) is a multivariate statistical method that can be used to analyze NMR data, revealing the inherent differences in serum metabolite levels in SD rats from the BC, SO, and ST groups. The score plot (Figure 2) of the PCA-DA analysis revealed that the NC group was clearly separated from the SO group and the ST group (*R*2 = 0.705, *Q*2 = 0.637), but the separation between the SO group and the ST group was not obvious.

### 2.3. Qualitative Analysis of Differentially Abundant Metabolites

The PCA-DA load plot can further identify the target metabolites that contribute to the separation phenomenon between the experimental groups in the score plot. The contribution of these metabolites to the separation phenomenon varies greatly and is generally determined based on their distance from the center point. Figure 3 shows that metabolites such as glucose, lactate, and pyruvate were farthest from the center point and made important contributions to the distinction between the experimental groups. That is, these three substances help distinguish between the BC group and the treated groups (SO and ST).

The variable importance in the projection (VIP) score plot can also identify metabolites that contribute to the separation between the experimental groups. A VIP value ≥ 1 for a target metabolite indicates that the metabolite plays an important role in distinguishing different experimental groups. Figure 4 shows that the metabolites with VIP values ≥ 1 were glucose, lactate, and pyruvate, which once again verified their importance in distinguishing between the BC group and the treated groups (SO and ST). The remaining metabolites with VIP values ≤ 1 made smaller contributions to distinguishing the relationships between the experimental groups. The red, yellow, and green boxes on the right side of Figure 4 represent the concentrations of the target metabolites in the different groups, with red representing high concentrations, yellow representing medium concentrations, and green representing low concentrations. Among the three metabolites with VIP values ≥ 1, lactate and pyruvate concentrations gradually increased in the BC group, SO group, and ST group, and the concentration of glucose gradually decreased in these groups.

### 2.4. Quantitative Analysis of Differentially Abundant Metabolites

Furthermore, we conducted a concentration analysis of the three metabolites, glucose, lactate, and pyruvate, that had important contributions to distinguishing the experimental groups using *t* tests (Figure 5). We found that the concentrations of lactate and pyruvate gradually increased, and the concentration of glucose gradually decreased in the NC, SO, and ST groups. These findings are consistent with the results of the VIP score plot analysis. Additionally, the concentration of glucose in the NC group was significantly greater than that in the SO and ST groups (*p* < 0.05), and there was no difference in the glucose concentrations between the SO and ST groups (*p* > 0.05). The concentrations of lactate and pyruvate in the NC group were significantly lower than those in the SO and ST groups (*p* < 0.05), and there was no difference in the lactate and pyruvate concentrations between the SO and ST groups (*p* > 0.05).

### 2.5. Analysis of the Affected Metabolic Pathways

In this study, metabolic pathway analysis was performed for the metabolites detected by ^1^H NMR in three experimental groups to identify significantly affected metabolic pathways, and a metabolic pathway analysis diagram (Figure 6) was generated. Each circle in the figure indicates a differential metabolic pathway obtained through metabolic pathway analysis, and the color of each circle indicates the *p* value. The redder the color is, the smaller the *p* value and the greater the significance. The size of the circle indicates the influence of the metabolic pathway, and the larger the circle is, the greater its influence. The closer the circle position is to the upper right diagonal, the more likely it is that the metabolic pathway was the differential pathway being sought. As shown in Figure 6, these pathways included the tricarboxylic acid cycle, pyruvate metabolism, glycine, serine, and threonine metabolism, alanine, aspartate, and glutamate metabolism.

## 3. Discussion

### 3.1. Changes in the Serum Metabolites of SD Rats Stung by Bees

The differentially abundant serum metabolites between the NC group and the treated groups included four amino acids (aspartic acid, glutamic acid, glutamine, and valine) and four organic acids (ascorbic acid, lactic acid, malonic acid, and pyruvic acid). Furthermore, these differentially abundant metabolites only showed significant differences between the NC group and the treated groups (*p* < 0.05), and no significant differences were detected between the treated groups (*p* > 0.05).

Amino acids play a key role as the building blocks of peptides, proteins, and phospholipids, and the serum levels of free amino acids can provide information about metabolic disturbances or specific metabolic processes [12]. In this study, the concentrations of glutamate and glutamine in the serum significantly increased after bee stings, which was particularly noteworthy. Glutamate has been previously described as the most abundant excitatory neurotransmitter [18]. It participates in energy processes by entering the Krebs cycle and in the biosynthesis of γ-aminobutyric acid (GABA), which is an inhibitory mediator in the brain [19,20]. Therefore, high levels of glutamate have been associated with headaches and other neurological diseases. The higher glutamate levels in the serum of rats stung by bees may be related to the body’s stress response and stimulation of the nervous system. However, the concentration of serum glutamine decreased after bee stings. This may be due to the breakdown of glutamine by glutaminase, which produces ammonia [21]. Additionally, the decrease in glutamine levels could be explained by the decrease in buffering capacity of the body due to the activity of bee venom. Glutamine is well known as a buffering amino acid used as a nitrogen carrier in biological fluids [22], leading to a detoxification process in which nitrogen from amino acid metabolism enters the urea cycle in the liver. Therefore, the decrease in the serum glutamine concentration after a bee sting may indicate its involvement in the detoxification of bee venom that spreads throughout the body.

Organic acids are acidic organic compounds (excluding amino acids) containing carboxyl groups that are widely present in organisms [23,24,25]. Among them, ascorbic acid greatly enhances the immune system, which can promote lymphocyte proliferation and rapidly consume ascorbic acid during infection [26]. The significant increase in the concentration of ascorbic acid in the serum of rats stung by bees may be an immune stress response to strong external stimuli. Serum lactate, as the most sensitive indicator reflecting tissue perfusion and oxygenation status, has been increasingly used in recent years to determine the severity and prognosis of diseases [27]. The increase in serum lactate levels in rats stung by bees was consistent with the acute and severe increase in lactate levels, which was a pathological increase. In addition, the serum pyruvate concentration of rats stung by bees also increased, and this effect can be observed in diabetes, heart failure, diarrhea, severe liver injury, acute infection, etc. Pyruvate is a product of glycolysis, and the blood lactate/pyruvate ratio is usually maintained at a fixed ratio. When the body is in an oxygen-deficient state, pyruvate is reduced to lactic acid, and a greater lactate/pyruvate ratio corresponds to more severe hypoxia [28]. The increase in this ratio in this study can be used to predict the severity of circulatory failure. A slight increase in activity will cause both lactate and pyruvate to increase simultaneously, but the ratio will remain unchanged.

### 3.2. The Impact of Slight Bee Stings on the Body

Most studies on bee sting injuries have focused on the allergic and toxic reactions caused by a single bee sting [15,16,17], while few studies have evaluated the body’s health after receiving many bee stings in a short time. Due to the restrictions of animal welfare laws, this experiment implemented one and two bee stings on SD rats to investigate the impact of the number of bee stings on the body. There was no difference between the treated groups, but there were differences between these groups and the NC group, indicating that the impacts of one and two bee stings on serum metabolism were similar. A small number of bee stings may be minor sting events for the body, and their impact on the body is relatively small and not easy to distinguish. Some reports have also suggested that the number of bee stings or the amount of bee venom that the body can tolerate in a short time is closely related to the body’s constitution, weight, underlying diseases, etc. [29,30]. In this study, the use of experimental rats may serve as a meaningful reference for evaluating human responses in terms of disease progression. For example, reports of patients who received 30 to 50 bee stings in a short time and received timely treatment had good prognoses. However, for patients with a history of severe allergies to bee venom, even one bee sting may cause a fatal allergic reaction. Therefore, in medical treatment, it is crucial to determine the time between bee sting occurrence and drug treatment and the time needed to remove the sting in a timely manner to determine patient treatment and prognosis [31]. When systemic toxic effects are expected, prognostic outcomes can be improved through proactive treatment. This study analyzed the impact of this kind of slight sting on the body and its underlying mechanisms through metabolomics technology, which has practical significance. This study helps to lay a foundation for future bee sting epidemiological research and medical treatment.

### 3.3. Effect of Bee Stings on Metabolic Pathways in the Body

Using PCA-DA and VIP analysis methods, we identified that glucose, lactate, and pyruvate were helpful for distinguishing between stung and non-stung samples. Lactate and pyruvate were discussed in detail above (Section 4.1), and this section further discusses glucose. Glucose in the blood, also known as blood sugar, is an important indicator of health [32]. The results of this study showed that bee stings caused a significant decrease in the serum glucose concentration in SD rats. Previous studies have confirmed that this may be caused by the action of melittin on multiple metabolic pathways in the body. For example, both melittin and bee venom can significantly reduce the levels of serum glucose and lipids, mainly by depolarizing pancreatic γ cells to increase insulin secretion [33,34]. In the treatment of diabetes, it is very important to increase insulin secretion and lipid regulatory mechanisms to reduce blood sugar levels by developing and utilizing various drugs, including animal and plant toxins [35,36,37]. In this study, bee stings significantly reduced serum glucose levels. Bee venom peptides possibly reduce blood glucose levels by increasing insulin secretion from pancreatic γ cells and promoting glucose uptake. Therefore, bee venom may be an option for treating or alleviating diabetes in the future.

Through ^1^H NMR technology, a total of 64 differentially abundant metabolites, including 22 amino acids and 20 organic acids, were detected in the serum of rats stung by bees. We conducted pathway analysis based on these metabolites and found that the most significantly affected pathways included the tricarboxylic acid cycle, pyruvate metabolism, glycine, serine, and threonine metabolism, alanine, aspartate, and glutamate metabolism, etc. This result was consistent with most reports. For example, Alonezi et al. studied the effect of major components of bee venom melittin on ovarian cancer cell metabolism and showed that melittin plays an important regulatory role in the tricarboxylic acid cycle, oxidative phosphorylation, purine and pyrimidine metabolism, and arginine/proline metabolism in the body [13]. The tricarboxylic acid cycle is a metabolic pathway that is common in aerobic organisms and serves as a hub connecting the three major metabolic pathways in the body [38]. Pyruvate also plays an important role as a hub connecting the metabolism of the three major nutrients. Therefore, bee stings caused damage to the tricarboxylic acid cycle and pyruvate metabolism, which not only affected energy metabolism but also affected material metabolism. The metabolic pathways affected in this study included glycine, serine, and threonine metabolism and alanine, aspartate, and glutamate metabolism. Alterations in these metabolic pathways of amino acids and organic acids lead to significant differences in amino acid and organic acid levels in serum and cause corresponding physiological or pathological changes. Amino acids and organic acids play vital roles as key components of proteins, carbohydrates, phospholipids, etc. [39]. They can also be used as precursor (biogenic amine, glucose, and heme) substances, neurotransmitters, and amino donors, participating in many aspects of cell function, including cell signaling, gene expression, and the regulation of amino acid self-transport [40,41].

In summary, we revealed the changes in serum metabolites and metabolic pathways induced by bee sting stimulation using metabolomics technology. However, the underlying mechanism has not yet been fully elucidated and requires further functional testing, biological validation, and other in-depth studies.

## 4. Materials and Methods

### 4.1. Animal Experiments, Sample Collection, and Preparation

This study used a total of 24 healthy male Sprague–Dawley (SD) rats provided by the Hubei Provincial Center for Disease Control and Prevention (Wuhan, China). Prior to the bee stings, all the experimental SD rats were individually housed in dedicated cages. The temperature was controlled at 25~28 °C, the humidity was controlled at 50~60%, and the light–dark cycle was 12 h. All SD rats were acclimated for 2 weeks before the experiment. In this study, all animal housing and experimental procedures followed the guidelines for the care and use of laboratory animals from the National Institutes of Health in the United States, and the experimental protocol was approved by the animal ethics committee (AEC code: 2023004) of Institute of Apicultural Research, Chinese Academy of Agricultural Sciences. The rats in this study were conducted at the animal department of Pharmaron Beijing Ltd. (Beijing, China). Prior to the experiments, all of the rats were acclimated in the same condition for at least 2 weeks. The SD rats had a body weight of 250 ± 10 g and were randomly divided into 3 groups (NC, SO, and ST, with 8 rats per group), as shown in Table 3. This experiment simulated the stinging events of a forager Italian honeybee (*Apis mellifera ligustica* Spin) colony under natural conditions in the field. The probability of aggressive behavior toward humans is relatively high as forager bees go out frequently, so a total of 30 forager bees (carrying pollen on their legs) were randomly collected at the entrance of the *Apis mellifera* colony hive to sting the SD rats. The control rats were treated exactly with the treatment group except stung by honeybees.

According to Table 3, the SD rats in the SO and ST groups were stung by bees. First, the dorsal hair of the SD rats was shaved close to the tail. After the shaving was complete, the SD rats were fixed in a dedicated rat holder, and the bees’ thorax was gently grasped with tweezers to position the stinger close to the shaved area of the SD rats for stinging. The success of the sting treatment was judged by whether bee stingers remained on the backs of the SD rats. In the SO group, each SD rat received only 1 bee sting, and in the ST group, each SD rat received 2 bee stings. The NC group served as the experimental control group, and the NC rats in this group received the same treatment as the SO and ST groups except stung by honeybees. Previous studies have demonstrated that after 3 h of venom stimulation, SD rats develop skin redness, depression, lethargy, drinking water, back licking, and self-grooming [42,43]. Three hours after the bee sting, the SD rats were anesthetized with carbon dioxide, and blood samples were collected from the carotid artery of each SD rat in each group (10 milliliters per rat). The blood was stored separately in blood cryopreservation tubes. The blood samples were centrifuged at 5000× *g* and 4 °C for 10 min. They were then incubated at 4 °C for 30 min and centrifuged again at 5000× *g* and 4 °C for 10 min. The supernatant (serum samples) from each sample was collected and stored in a −80 °C freezer for future ^1^H NMR analysis [43].

Before NMR analysis [43,44], the frozen serum sample was first thawed and vortexed for 30 s. After mixing, the serum sample was centrifuged at 13,000× *g* rpm for 2 min at 4 °C, and the supernatant was filtered using a 3 kDa microfilter. Approximately 450 μL of the filtered solution was transferred to a clean centrifuge tube, which had been prefilled with 50 μL of anachro-certified 2,2-dimethyl-2-silapentane-5-sulfonate (DSS) standard solution (ACDSS, 4.136 mM). ACDSS served as an internal standard compound for NMR analysis, with a chemical shift of 0.0 ppm in the NMR spectrum [43,45].

### 4.2. ^1^H NMR Spectroscopy Analysis

The metabolites in the serum samples were detected on a nuclear magnetic resonance spectrometer with the specific parameters of a Bruker AV III 600 MHz (Bruker Biospin, Milton, ON, Canada) [19]. The instrument was equipped with a reverse freezer with a working parameter of 600.13 MHz. The ^1^H NMR spectra of serum samples were generated by standard NOESY (noesygppr1d.comp; Bruker Biospin) pulse sequence using 64 scans, 32,786 data points, and a spectral width of 8000 Hz.

### 4.3. Data Analysis

For raw data pre-processing, NMR spectra were acquired with a 12 ppm sweep width, 4 dummy scans, 4 s acquisition line, and 32 transients. Then, all spectra were zero-filled to 128,000 data points, Fourier-transformed with a 0.5 Hz line broadly applied, and manually phased and baseline-corrected using VNMR software (version 6.1C, Varian Inc., Palo Alto, California, USA). The ^1^H NMR free induction decay (FID) signal was imported into the Chenomx NMR suite software (version 8.0, Chenomx, Edmonton, AB, Canada) for automatic Fourier transformation, phase adjustment, and baseline correction. The DSS-d6 peak (0.0 ppm) was used as the standard for all spectral chemical shifts, and it was subjected to an inversion convolution operation to adjust the spectral peak shape (CSI). Based on the relevant information (such as chemical shift, peak shape, half-peak width, and coupling splitting) of the ^1^H NMR spectrum, the concentration and spectral peak area of DSS-d6 were used as the standard, and the signals of the serum sample spectra were compared and analyzed one by one in combination with Chenomx’s built-in database. Finally, the metabolites and corresponding absolute concentration values were obtained [18,19].

Metabolite set enrichment analysis (MSEA) and metabolic pathway analysis (Appendix A) were performed using Metaboanalyst 3.0 (https://www.metaboanalyst.ca/faces/ModuleView.xhtml, accessed on 6 July 2016). Data were expressed as mean ± standard error of the mean and plotted as bar graphs using GraphPad prim software (version 8, GraphPad, San Diego, CA, USA). Differences between groups were analyzed using one-way ANOVA and Duncan’s post hoc test (*p* < 0.05) in SPSS software (version 25.0, SPSS Inc., Chicago, IL, USA).

## Figures and Tables

**Figure 1 ijms-25-06365-f001:**
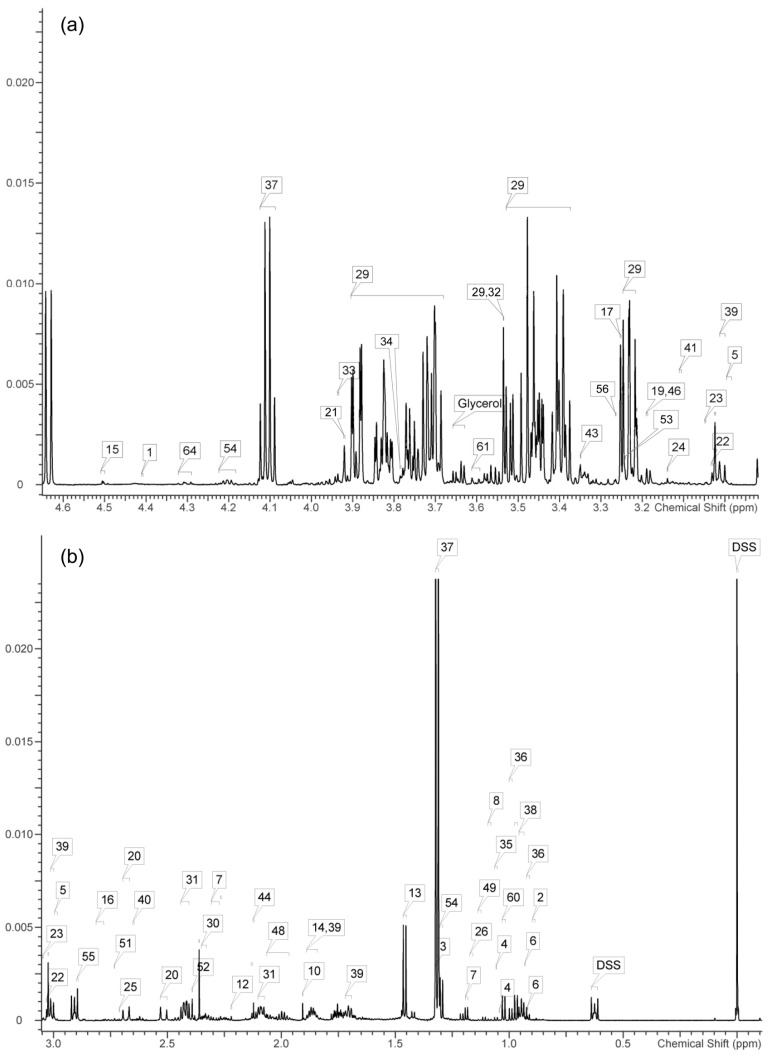
^1^H NMR spectra of serum samples from rats stung one time (**a**) and non-stung (**b**) by honeybees.

**Figure 2 ijms-25-06365-f002:**
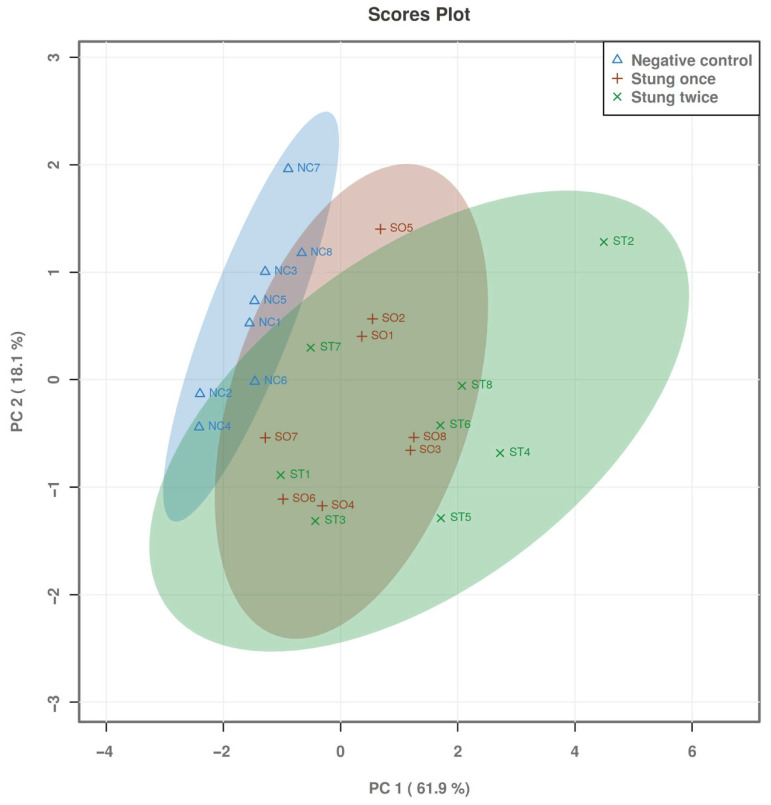
Principal Component Analysis (PCA-DA) score plot of serum metabolites in SD rats stung by honeybees. The separation degree of the SO group and ST group was weak, and the separation degrees of the SO group and BC group and the ST group and BC group were strong.

**Figure 3 ijms-25-06365-f003:**
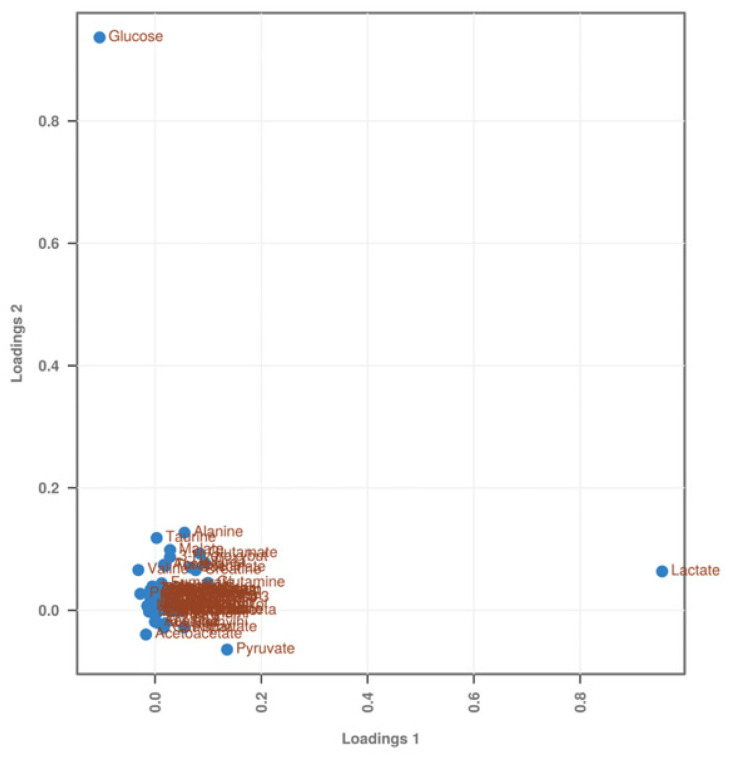
PCA-DA loading diagram of serum metabolites in SD rats stung by honeybees. Lactic acid, glucose, and pyruvic acid were far from the center of the substances and were identified as differentially abundant metabolites.

**Figure 4 ijms-25-06365-f004:**
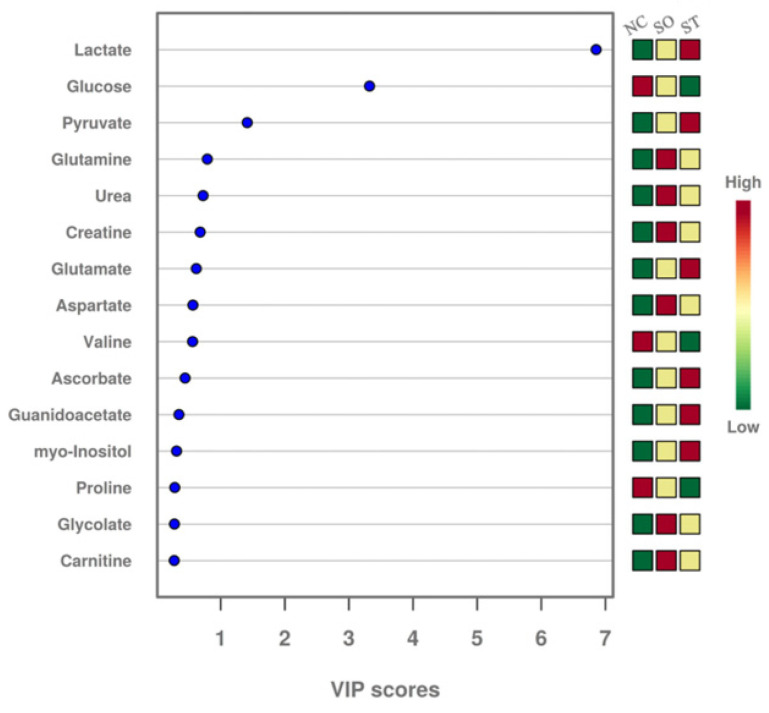
VIP score plot of serum metabolites in SD rats stung by honeybees. Lactic acid, glucose, and pyruvate had VIP values greater than 1 and were identified as differentially abundant metabolites.

**Figure 5 ijms-25-06365-f005:**
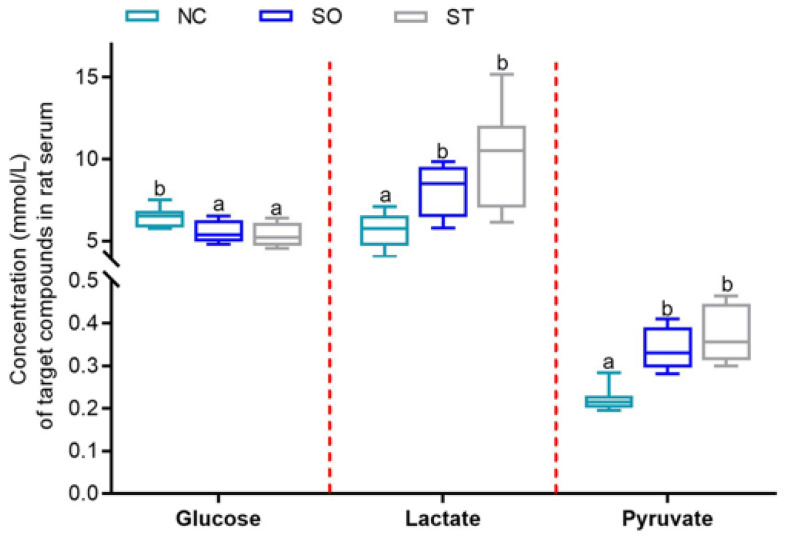
Quantitative analysis of serum differentially abundant metabolites based on ^1^H NMR spectra of serum samples. Different lowercase letters indicate significant differences at the *p* < 0.05.

**Figure 6 ijms-25-06365-f006:**
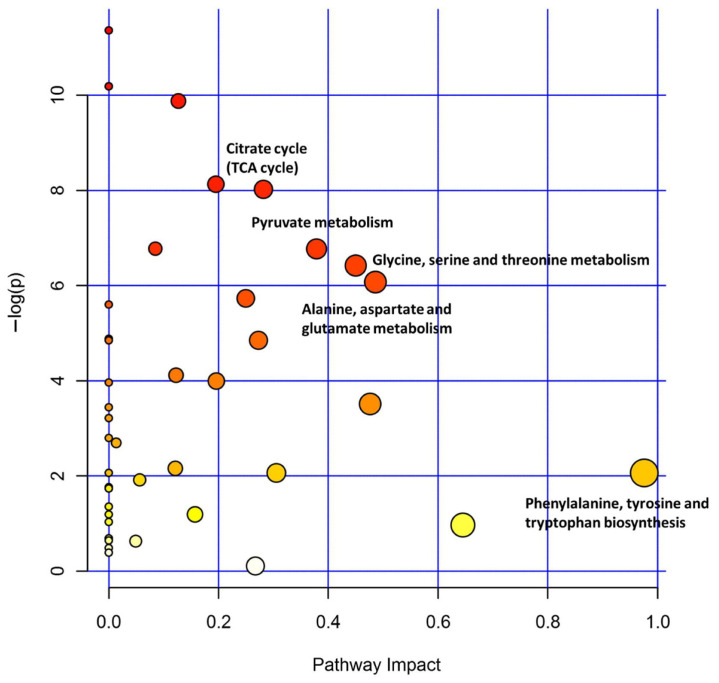
Quantitative analysis of serum metabolites based on ^1^H NMR spectra of serum samples. The darker and larger circles, which were also near the upper right corner, were screened as differential metabolic pathways and are marked in the figure.

**Table 1 ijms-25-06365-t001:** Details of the amino acids identified by ^1^H NMR.

Compound Labels of NMR Spectra	HMDB Accession Number	Amino Acids	Treatment Groups (*n* = 8)		
			NC	SO	ST
13	HMDB00161	L-Alanine	0.496 ± 0.094 ^a^	0.541 ± 0.095 ^a^	0.517 ± 0.079 ^a^
14	HMDB00517	L-Arginine	0.160 ± 0.038 ^a^	0.153 ± 0.029 ^a^	0.162 ± 0.021 ^a^
16	HMDB00191	L-Aspartic acid	0.024 ± 0.005 ^a^	0.048 ± 0.007 ^b^	0.047 ± 0.008 ^b^
17	HMDB00043	Betaine	0.135 ± 0.026 ^a^	0.138 ± 0.022 ^a^	0.127 ± 0.024 ^a^
18	HMDB00062	L-Carnitine	0.051 ± 0.014 ^a^	0.068 ± 0.022 ^a^	0.063 ± 0.014 ^a^
31	HMDB00641	L-Glutamine	0.454 ± 0.050 ^b^	0.315 ± 0.051 ^a^	0.316 ± 0.017 ^a^
32	HMDB00123	Glycine	0.235 ± 0.071 ^a^	0.245 ± 0.063 ^a^	0.242 ± 0.042 ^a^
36	HMDB00172	L-Isoleucine	0.075 ± 0.010 ^a^	0.072 ± 0.008 ^a^	0.066 ± 0.012 ^a^
38	HMDB00687	L-Leucine	0.075 ± 0.022 ^a^	0.072 ± 0.013 ^a^	0.067 ± 0.020 ^a^
39	HMDB00182	L-Lysine	0.264 ± 0.031 ^a^	0.239 ± 0.047 ^a^	0.276 ± 0.039 ^a^
44	HMDB00696	L-Methionine	0.047 ± 0.007 ^a^	0.051 ± 0.006 ^a^	0.049 ± 0.005 ^a^
45	HMDB00201	L-Acetylcarnitine	0.016 ± 0.003 ^a^	0.021 ± 0.009 ^a^	0.019 ± 0.005 ^a^
47	HMDB00159	L-Phenylalanine	0.042 ± 0.005 ^a^	0.049 ± 0.011 ^a^	0.048 ± 0.005 ^a^
48	HMDB00162	L-Proline	0.229 ± 0.040 ^a^	0.223 ± 0.027 ^a^	0.207 ± 0.041 ^a^
51	HMDB00271	Sarcosine	0.004 ± 0.001 ^a^	0.003 ± 0.000 ^a^	0.003 ± 0.001 ^a^
53	HMDB00251	Taurine	0.033 ± 0.048 ^a^	0.080 ± 0.153 ^a^	0.012 ± 0.034 ^a^
54	HMDB00167	L-Threonine	0.230 ± 0.054 ^a^	0.236 ± 0.057 ^a^	0.216 ± 0.051 ^a^
57	HMDB00158	L-Tyrosine	0.083 ± 0.016 ^a^	0.084 ± 0.012 ^a^	0.072 ± 0.012 ^a^
60	HMDB00883	L-Valine	0.167 ± 0.022 ^b^	0.139 ± 0.016 ^ab^	0.120 ± 0.015 ^a^
63	HMDB00725	4-Hydroxyproline	0.057 ± 0.014 ^a^	0.055 ± 0.013 ^a^	0.057 ± 0.013 ^a^

Note: Negative control (NC), stung 1 time (SO), stung 2 times (ST); different lowercase letters represent significant differences (*p* < 0.05).

**Table 2 ijms-25-06365-t002:** Details of the organic acids identified by ^1^H NMR.

Compound Labels of NMR Spectra	HMDB Accession Number	Organic Acids	Treatment Groups (*n* = 8)		
			NC	SO	ST
2	HMDB00008	2-Hydroxybutyric acid	0.007 ± 0.002 ^a^	0.006 ± 0.002 ^a^	0.006 ± 0.004 ^a^
3	HMDB00729	Alpha-Hydroxyisobutyric acid	0.003 ± 0.001 ^a^	0.004 ± 0.002 ^a^	0.005 ± 0.001 ^a^
4	HMDB00005	2-Ketobutyric acid	0.003 ± 0.002 ^a^	0.002 ± 0.001 ^a^	0.005 ± 0.005 ^a^
5	HMDB00208	Oxoglutaric acid	0.031 ± 0.007 ^a^	0.039 ± 0.009 ^a^	0.033 ± 0.008 ^a^
6	HMDB00695	Ketoleucine	0.006 ± 0.002 ^a^	0.005 ± 0.002 ^a^	0.005 ± 0.002 ^a^
7	HMDB00357	3-Hydroxybutyric acid	0.067 ± 0.018 ^a^	0.063 ± 0.017 ^a^	0.064 ± 0.030 ^a^
8	HMDB00491	3-Methyl-2-oxovaleric acid	0.005 ± 0.002 ^a^	0.004 ± 0.001 ^a^	0.004 ± 0.001 ^a^
10	HMDB00042	Acetic acid	0.071 ± 0.017 ^a^	0.056 ± 0.013 ^a^	0.067 ± 0.029 ^a^
11	HMDB00060	Acetoacetic acid	0.021 ± 0.008 ^a^	0.018 ± 0.009 ^a^	0.023 ± 0.010 ^a^
15	HMDB00044	Ascorbic acid	0.038 ± 0.010 ^a^	0.075 ± 0.012 ^b^	0.077 ± 0.008 ^b^
20	HMDB00094	Citric acid	0.131 ± 0.033 ^a^	0.137 ± 0.023 ^a^	0.134 ± 0.027 ^a^
27	HMDB00142	Formic acid	0.032 ± 0.006 ^a^	0.034 ± 0.005 ^a^	0.030 ± 0.003 ^a^
28	HMDB00134	Fumaric acid	0.004 ± 0.002 ^a^	0.006 ± 0.007 ^a^	0.005 ± 0.003 ^a^
33	HMDB00115	Glycolic acid	0.026 ± 0.014 ^a^	0.038 ± 0.014 ^a^	0.037 ± 0.012 ^a^
35	HMDB01873	Isobutyric acid	0.008 ± 0.002 ^a^	0.006 ± 0.001 ^a^	0.006 ± 0.001 ^a^
37	HMDB00190	L-Lactic acid	5.698 ± 1.044 ^a^	9.251 ± 1.332 ^b^	10.345 ± 2.147 ^b^
41	HMDB00691	Malonic acid/Malonate	0.008 ± 0.002 ^a^	0.022 ± 0.004 ^b^	0.020 ± 0.004 ^b^
50	HMDB00243	Pyruvic acid/Pyruvate	0.221 ± 0.028 ^a^	0.340 ± 0.052 ^b^	0.371 ± 0.067 ^b^
52	HMDB00254	Succinic acid/Succinate	0.022 ± 0.010 ^a^	0.030 ± 0.017 ^a^	0.039 ± 0.040 ^a^

Note: Negative control (NC), stung 1 time (SO), stung 2 times (ST); different lowercase letters represent significant differences (*p* < 0.05).

**Table 3 ijms-25-06365-t003:** Treatments of SD rats in each group.

Groups	Quantity of SD Rats	Treatments at 0 h	Blood Collection Time (h)
NC	8	Negative control	3
SO	8	Sting one time	3
ST	8	Sting two times	3

## Data Availability

The original contributions presented in this study are included in the article; further inquiries can be directed to the corresponding authors.

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
