# Peer review of "Being Stung Once or Twice by Bees (Apis mellifera L.) Slightly Disturbed the Serum Metabolome of SD Rats to a Similar Extent"

_ijms, 2024, doi:10.3390/ijms25126365_

Round 1

Reviewer 1 Report

Comments and Suggestions for Authors

The paper describes an NMR approach to analyze rats serum metabolome and understand the impact of 1 and 2 bee stings.

The paper reads well and could be interesting but lacks statistical robustness, comments on its limitation and could benefit some extra analysis and improvement of figures.

Comments:

Line 32: "re-cent" correct to "recent"

Figure 1: impossible to see well, too small and bad quality and no reference to all the numbers reported.

Analysis: PLS-DA is fundamentally biased as it is a supervised technique, so PLS-DA scoreplots are not a robust way to show real differences. Did the author try a PCA? That would be the first method to try and describe as it is unsupervised. HCA or ASCA would also be better techniques to show. With limited number of samples the PLS-DA will be mostly overfitting.

Is the concentration analysis just done on Chenomx concentrations? Those can be highly miscalculated. Did you try confirming it with different methods?

Some of the assigned metabolites reported as significantly changing are in regions highly overlapping with other stuff. For example, are you sure about ascorbate? That will be deeply affected by glucose peaks. Also glutamine and glutamate have major overlaps, as it is not clear from Fig.1 which peaks are used this can be very misleading. A full table, even supplementary, reporting all peaks is fundamental.

There is no information in the methods on the pulse sequence used and its details.

The study fundamentally lacks a criticism of its limitation: low n. of rats, smaller bodies than humans, etc. Moreover, were the control rats also shaved and kept in a rat holder? Differences in stress levels may also just be the reason of some of the metabolic changes if the controls were just treated differently and this is not sufficiently clear.

Comments on the Quality of English Language

Paper reads fine.

Author Response

Dear Editor and Reviewer:
Thank you for your letter and for the reviewer’ comments concerning our manuscript entitled “Being stung once or twice by bees (Apis mellifera L.) slightly disturbed the serum metabolome of SD rats to a similar extent” (ijms-2992134). Those comments are all valuable and very helpful in improving the quality of our manuscript. We have carefully studied the comments and made corrections which we hope to get approval. The revised portions are marked with “track changes” in our updated manuscript. The main corrections in this article and the responses to the reviewer’s comments are as flows:

Responses to the comments of reviewer #1:

1.Response to comment: (Figure 1: impossible to see well, too small and bad quality and no reference to all the numbers reported.)

Response: Thank you for your suggestion. We have changed the Figure1 with high quality one.

2.Response to comment: (Analysis: PLS-DA is fundamentally biased as it is a supervised technique, so PLS-DA scoreplots are not a robust way to show real differences. Did the author try a PCA? That would be the first method to try and describe as it is unsupervised. HCA or ASCA would also be better techniques to show. With limited number of samples the PLS-DA will be mostly overfitting. )

Response: We thank the reviewer for their observation. We have carefully checked the original data and find that the Figure2 and Figure3 are actually generated by Principal Component Analysis (PCA-DA). We have replaced the Figure2 and Figure3 with the PCA-DA data in our updated manuscript and attached the PLS-DA data here (Response Figure1). We are so sorry for this mistake.

Response Figure1. (A-B). PLS-DA score plot and PLS-DA loading diagram of serum metabolites in SD rats stung by honeybees.

3.Response to comment: (Is the concentration analysis just done on Chenomx concentrations? Those can be highly miscalculated. Did you try confirming it with different methods?)

Response: Thank you for your valuable comments. We have actually confirmed our data with RNA-Seq. The transcriptomic result shown that it has highly overlapped signaling pathways. We are submitting our RNA-Seq data to another journal and it is not convenient to present the results here.

4.Response to comment: (Some of the assigned metabolites reported as significantly changing are in regions highly overlapping with other stuff. For example, are you sure about ascorbate? That will be deeply affected by glucose peaks. Also glutamine and glutamate have major overlaps, as it is not clear from Fig.1 which peaks are used this can be very misleading. A full table, even supplementary, reporting all peaks is fundamental. )

Response: Thank you for your suggestion. Indeed, we should use standard substance to confirm the compounds which are overlapping with other elements. We have carefully check out data and make sure all of the 1H NMR spectra of standard compounds contained in the Chenomx library were identified and quantified by the peak area and concentration of DSS-d6. We have add the compounds lable numbers of NMR spectra in Table 1 and Table 2 and made a full table which reports all peaks of 1H NMR spectra. Please check the Table S1.

5.Response to comment: (There is no information in the methods on the pulse sequence used and its details.)

Response: We have made correction according to the reviewer’s comments. All information of pulse sequence used in this research have been added to method part in our updated manuscript. The detail information is: The metabolites in the serum samples were detected on a nuclear magnetic resonance spectrometer with the specific parameters of a Bruker AV III 600 MHz (Bruker Biospin, Milton, Canada). The instrument was equipped with a reverse freezer with a working parameter of 600.13 MHz. The 1H NMR spectra of serum samples were generated by standard NOESY (noesygppr1d.comp; Bruker BioSpin) pulse sequence using 64 scans, 32,786 data points, a spectral width of 8,000 Hz.

6.Response to comment: (The study fundamentally lacks a criticism of its limitation: low n. of rats, smaller bodies than humans, etc. Moreover, were the control rats also shaved and kept in a rat holder? Differences in stress levels may also just be the reason of some of the metabolic changes if the controls were just treated differently and this is not sufficiently clear.)

Response: Thank you for your question. Indeed, some of studies used 25 rats/group for their 1H NMR serum metabolomics research, however, there are also a lot of studies used 6 or 10 rats/group [1,2]. We used 8 rats/group in this research, the agglomeration of our PLS-DA scores show the repetition of our rat samples. We believed that 8 rats per group are enough to against individual variations. It's for sure that all the control rats were shaved and kept in the same rat holder with the SD rats stung by honeybees. The control rats were treated exactly with the treatment group except stung by honeybees. We have updated our manuscript about the detail information the animals we used.

References

  1. Feng, Q. S.; Tong, L.; Lu, Q.; Liu, S.; Zhao, L. S.; Xiong, Z. L., 1H NMR serum metabolomics and its endogenous network pharmacological analysis of Gushudan on kidney-yang-deficiency-syndrome rats. Anal Biochem 2022, 643,114580.
  2. Kim, J. A.; Choi, H. J.; Kwon, Y. K.; Ryu, D. H.; Kwon, T. H.; Hwang, G. S., 1H NMR-Based metabolite profiling of plasma in a Rat model of chronic kidney disease. Plos One 2014, 9, e85445.

Reviewer 2 Report

Comments and Suggestions for Authors

The present investigation describes the serum metabolomic profile of Sprague-Dawley rats exposed to one and two Apis mellifera bee stings, compared to rats without stings (negative control). One-dimensional proton nuclear magnetic resonance spectroscopy (1H NMR) was used as the analytical platform. The work is interesting and is described clearly and concisely. It seems to me that the experimental part is well conducted, but relevant experimental information was omitted. The research topic and the results obtained fulfill the requirements to eventually be published in the International Journal of Molecular Sciences. Nevertheless, the authors are encouraged to consider the following observations and recommendations:

The standard approach in multivariate statistical analysis typically involves the initial application of principal component analysis (PCA). In this case, it would be valuable to clarify the reasons behind excluding this method from further consideration.

From a phylogenetic point of view, Apis mellifera have been classified according to lineages or DNA types. Accordingly, the authors should specify, if possible, the subspecies of Apis mellifera used to sting the SD rats.

Is there a relationship between the symptoms and the composition of the honey venom?

The Chenomx program allows the generation or development of a spectrum containing the proportional sum of the metabolites identified from the samples studied. This provides valuable information for comparison with the spectra obtained from the analyzed samples. Authors can take advantage of the capabilities offered by the Chenomx program.

Line 90, 91: In the 1H NMR, place "1" in the appropriate position (superscrib).

The authors omitted pertinent information regarding the acquisition parameters of the 1H NMR spectra. For instance:

1.      It is inferred that a mixture of H2O and D2O was utilized for the 1H NMR sample measurements, with the inclusion of D2O serving to track the deuterium signal. However, it is essential to explicitly mention this information.

2.      Binning size?, spectral window?, acquisition time?, number of scans?, gain? Number of complex data points collected? Sequence pulse used for residual H2O presaturation?, etc.

3.      Did the authors use a quality control sample? Please, explain.

Author Response

Dear Editor and Reviewer:
Thank you for your letter and for the reviewer’ comments concerning our manuscript entitled “Being stung once or twice by bees (Apis mellifera L.) slightly disturbed the serum metabolome of SD rats to a similar extent” (ijms-2992134). Those comments are all valuable and very helpful in improving the quality of our manuscript. We have carefully studied the comments and made corrections which we hope to get approval. The revised portions are marked with “track changes” in our updated manuscript. The main corrections in this article and the responses to the reviewer’s comments are as flows:

Responses to the comments of reviewer #2:

1.Response to comment: (The standard approach in multivariate statistical analysis typically involves the initial application of principal component analysis (PCA). In this case, it would be valuable to clarify the reasons behind excluding this method from further consideration.)

Response: We thank the reviewer for their observation. We have carefully checked the original data and find that the Figure2 and Figure3 are actually generated by Principal Component Analysis (PCA-DA). We have replaced the Figure2 and Figure3 with the PCA-DA data in our updated manuscript and attached the PLS-DA data here (Response Figure1). We are so sorry for this mistake.

Response Figure1. (A-B). PLS-DA score plot and PLS-DA loading diagram of serum metabolites in SD rats stung by honeybees.

2.Response to comment: (From a phylogenetic point of view, Apis mellifera have been classified according to lineages or DNA types. Accordingly, the authors should specify, if possible, the subspecies of Apis mellifera used to sting the SD rats.)

Response: Thank you for your valuable comments. We have added the detailed information about Apis mellifera in our updated manuscript, including the subspecies information of Apis mellifera we used in this study. The information of bees we used is: the Italian honey bee, Apis mellifera ligustica Spin

3.Response to comment: (Is there a relationship between the symptoms and the composition of the honey venom?)

Response: Thank you for your question. Indeed, many studies shown that there is a closely correlation between the symptoms and the composition of bee venom [3-5]. We mainly focus on the serum metabolome different of SD rats un-stung and stung (once or twice) by honeybees.

4.Response to comment: (The Chenomx program allows the generation or development of a spectrum containing the proportional sum of the metabolites identified from the samples studied. This provides valuable information for comparison with the spectra obtained from the analyzed samples. Authors can take advantage of the capabilities offered by the Chenomx program.)

Response: Thank you for your question. Exactly, we indeed used Chenomx NMR Suite software (version 8.0, Chenomx, Edmonton, Canada) to identify and quantify the metabolites. Please check “Data analysis” section in our manuscript.

5.Response to comment: Line 90, 91: In the 1H NMR, place "1" in the appropriate position (superscrib).
Response: We have made correction according to the reviewer’s comments.

6.Response to comment: (The authors omitted pertinent information regarding the acquisition parameters of the 1H NMR spectra. For instance:

(1). It is inferred that a mixture of H2O and D2O was utilized for the 1H NMR sample measurements, with the inclusion of D2O serving to track the deuterium signal. However, it is essential to explicitly mention this information.

(2). Binning size?, spectral window?, acquisition time?, number of scans?, gain? Number of complex data points collected? Sequence pulse used for residual H2O presaturation?, etc.

(3). Did the authors use a quality control sample? Please, explain.)

Response: Thank you for your question. We have made correction according to the reviewer’s comments about the detail information of the 1H NMR spectra we used in this study.

For the sub-question (1): All of the 1H NMR spectra of standard compounds contained in the Chenomx library were identified and quantified by the peak area and concentration of DSS-d6. We have made a full table which reports all peaks of 1H NMR spectra. Please check the Table S1.

For the sub-question (2): The 1H NMR spectra of serum samples were generated by standard NOESY (noesygppr1d.comp; Bruker BioSpin) pulse sequence using 64 scans, 32,786 data points, a spectral width of 8,000 Hz. For raw data pre-processing, NMR spectra were acquired with a 12 ppm sweep width, 4 s acquisition line, 4 dummy scans, and 32 transients. Then, all spectra were zero-filled to 128 000 data points, Fourier-transformed with a 0.5 Hz line broadly applied, and manually phased and baseline-corrected using VNMR software.

For the sub-question (3): We have checked our experimental record and find that all the control rats were shaved and kept in the same rat holder with the SD rats stung by honeybees. The control rats were treated exactly with the treatment group except stung by honeybees. We have updated our manuscript about the detail information the animals we used.

References

  1. Zheng, X.; Wang, X.; Wang, Q.Y.; Liu, M.Y.; Peng, W.J.; Zhao, Y.Z. Severe pathological changes in the blood and organs of SD rats stung by honeybees. Toxicon 2023, 231, 107196.
  2. De Roodt, A. R.; Lanari, L. C.; Lago, N. R.; Bustillo, S.; Litwin, S.; Morón-Goñi, F.; Gould, E. G.; van Grootheest, J. H.; Dokmetjian, J. C.; Dolab, J. A.; Irazú, L.; Damin, C. F., Toxicological study of bee venom (Apis mellifera mellifera) from different regions of the province of Buenos Aires, Argentina. Toxicon 2020, 188, 27-38.
  3. Uthawarapong, P.; Benbow, M. E.; Suwannapong, G., First study on the effect of Asiatic honey bee (Apis cerana) venom on cutaneous, hepatic and renal rat tissues. J Apicult Res 2019, 58, 764-771.